# Performance evaluation of survival regression models in analysing Swedish dental implant complication data with frailty

**Adeniyi Francis Fagbamigbe**[1,2,3]*, **Karolina Karlsson**[4], **Jan Derks**[4], **Max Petzold**[5]

**1** Department of Epidemiology and Medical Statistics, College of Medicine, University of Ibadan, Ibadan, Nigeria, **2** Division of Health Sciences, Populations, Evidence and Technologies Group, University of Warwick, Coventry, United Kingdom, **3** Division of Population and Behavioural Studies, School of Medicine, University of St Andrews, St Andrews, United Kingdom, **4** Department of Periodontology, Institute of Odontology, The Sahlgrenska Academy at University of Gothenburg, Gothenburg, Sweden, **5** School of Public Health and Community Medicine, Institute of Medicine, The Sahlgrenska Academy at University of Gothenburg, Gothenburg, Sweden

* franstel74@yahoo.com, fadeniyi@cartafrica.org

**Data Availability Statement:** The data underlying the results presented in the study are available from the regional Ethical Committee, Gothenburg, Sweden (Dnr 290-10), ClinicalTrials.gov This study is human research based on sensitive data which

## Abstract

The use of inappropriate methods for estimating the effects of covariates in survival data with frailty leads to erroneous conclusions in medical research. This study evaluated the performance of 13 survival regression models in assessing the factors associated with the timing of complications in implant-supported dental restorations in a Swedish cohort. Data were obtained from randomly selected cohort (n = 596) of Swedish patients provided with dental restorations supported in 2003. Patients were evaluated over 9 years of implant loss, peri-implantitis or technical complications. Best Model was identified using goodness, AIC and BIC. The loglikelihood, the AIC and BIC were consistently lower in flexible parametric model with frailty (df = 2) than other models. Adjusted hazard of implant complications was 45% (adjusted Hazard Ratio (aHR) = 1.449; 95% Confidence Interval (CI): 1.153–1.821, p = 0.001) higher among patients with periodontitis. While controlling for other variables, the hazard of implant complications was about 5 times (aHR = 4.641; 95% CI: 2.911–7.401, p<0.001) and 2 times (aHR = 2.338; 95% CI: 1.553–3.519, p<0.001) higher among patients with full- and partial-jaw restorations than those with single crowns. Flexible parametric survival model with frailty are the most suitable for modelling implant complications among the studied patients.

## Introduction

Survival regression methods are commonly used to explore heterogeneity among subjects in medical research [1] and to estimate prognostic factors for survival [2–6]. However, one of the major challenges in survival analysis modelling is clustering among followed subjects, otherwise known as frailty [7,8]. The concept of frailty is an issue of discourse in statistical modelling, including survival analysis. Frailty is a group-specific latent random effect that multiplies

cannot be publicly shared. Data contain potentially identifying and sensitive patient information. Interested researchers can contact the Swedish Ethical Review Authority or the Head of Institute, Peter Lingström: Swedish Ethical Review Authority (Etikprövningsmyndigheten) registrator@etikprovning.se Box 2110 750 02 Uppsala, Sweden +46 (0)10-475 08 00 Peter Lingström peter.lingstrom@odontologi.gu.se Professor of Cariology, Institute of Odontology, University of Gothenburg https://www.gu.se/en/ odontology/about-us/contact info@odontologi.gu. se +46 (0)31-786 2932.

**Funding:** No specific funding was received. However, the Consortium for Advanced Research and Training in Africa (CARTA) supported AFF to visit the University of Gothenburg as part of his fellowship at the University of Warwick. CARTA is jointly led by the African Population and Health Research Center and the University of the Witwatersrand and funded by the Carnegie Corporation of New York (Grant No–B 8606.R02), Sida (Grant No:54100029), and the DELTAS Africa Initiative (Grant No: 107768/Z/15/Z). The data analysis was carried out during the visit.

**Competing interests:** The authors have declared that no competing interests exist.

**Abbreviations:** AFT, Accelerated Failure Time; AIC, Akaike information criteria; aHR, adjusted Hazard Ratio; BIC, Bayesian information criteria; CI, Confidence Interval; CPH, Cox Proportional Hazard; FPSR, Flexible parametric survival regression; IQR, Interquartile range; PH, Proportional hazard; RP, Royston-Palmar.

into the hazard function. The frailties are unobservable positive quantities. They follow a gamma distribution with a mean of 1 and variance to be estimated from the data. Theoretically, any distribution with positive support (mean = 1) and finite variance may be used to model frailty. In most cases, shared-frailty models are used to model the within-group correlation. Observations within a particular group are often referred to as correlated because they share the same frailty. Let us consider the case of patients attending a dental clinic. Any of the patients may have issues with any of their teeth after a particular intervention. The teeth of a single patient form a cluster. While it is reasonable to assume independence of the dental patients, it is incorrect to assume that the occurrence of infection to any of the teeth within each patient is independent. It is necessary therefore to accommodate the potential "dependency" by assuming that it was the result of a latent patient-level effect or frailty. Non-consideration of clustering in clustered data causes poor model fit and biased estimates. This suggests that a mixed-effects model that contains both the random and fixed effects would be most appropriate to model such outcomes.

The alternative, traditional survival regression models, divided into parametric (Poisson, Weibull), semiparametric (Cox), and nonparametric (Kaplan–Meier) have distinct disadvantages that could make them unsuitable to correctly predict survival outcomes [1,9]. For instance, the Kaplan-Meier model does not accommodate covariates, hence its utilization is limited [1,5]. Although the Cox proportional hazard (CPH) model is the most commonly used model in survival analyses [1,10,11] and has been used extensively in the literature [4,11–13], its efficiency is limited for short observation periods [1]. Further, its distribution-free assumption is often violated in long-term studies. In either case, many of the subjects may not have experienced the event of interest and, thus, survival and cumulative hazard functions are incomplete and cannot be extrapolated in the CPH [1]. The CPH models assume a constant hazard, an assumption that is also frequently violated [14,15]. The Cox model has an advantage in that it does not assume the form of the baseline hazard function, therefore, not hindering the prediction of hazards and other related functions for a given set of covariates but this advantage gave birth to its major disadvantage [14]. Moreover, survival and cumulative hazard functions of the CPH model are step functions and, thus, limit the possibility of having smooth functions [10,16].

Parametric models, such as the exponential and Weibull models [1], attempted to overcome some of the shortcomings of the CPH model by producing smooth predictions by assuming a functional form of the hazard [1,17] and directly estimating the absolute and relative effects [14]. The models can be used to estimate the smooth cumulative hazard functions and hazard ratios of risk factors and extrapolate survival and cumulative functions [1]. Nonetheless, the models assume that the survival and hazard functions have a specific distribution which is often too structured and sometimes unrealistic for use with real data [9,10]. In addition, parametric models with complex underlying hazard fail to capture true effects [18,19]. Thus, in most cases, parametric models have insufficient flexibility and, thereby, produce biased cumulative hazard and survival functions [10].

While the disadvantages of non-parametric models can be overcome by the use of stratification, the number of factors used for such stratification may be limited [1]. Another way of alleviating the challenges of the CPH is to use a sufficiently large sample size and extensive study duration [20]. Also, parametric survival models may be useful if available data do not violate the underlying assumptions of the distributions. Despite these mitigations, none of the non-parametric, semi-parametric or parametric models is flexible enough to accommodate structural composition of all real-life data.

Royston and Parmar (RP) therefore developed flexible parametric survival regression (FPSR) models as a result of lack of adequate flexibility of the Cox and parametric survival

models [9,10,21]. The FPSR model offers a compromise between the CPH and parametric models and retains the desired features of both types of models. The flexible parametric approach works by relaxing the assumption of linearity of log time by using restricted cubic splines [10,22]. The overall advantages of the FPSR models have been reported in the literature [1,9,14,18,19,23,24]. CPH, parametric and the FPSR models have incorporated frailty options.

Different factors could be associated with complications affecting dental restorations supported by implants but it is not known how non-consideration of the clustering nature of the implants (multiple implants in one subject) affect outcomes of the modelling approach. The need for appropriate statistical models for accurate medical inferences and decisions motivated this study. It is designed to evaluate and compare the application of CPH, parametric and FPSR models to complications affecting dental implants. Implants are clustered within patients and intra-cluster dependency may occur. We hypothesized that models with frailty, and the flexible parametric model with frailty, in particular, would perform better than all other range of models. We aimed to apply different survival analysis regression models to a dataset originating from Sweden [25,26] and assess the performance of the models to identify the model with the best data fit. We considered the (i) Cox proportional hazard models for frailty, (ii) Multilevel mixed-effects parametric survival models for proportional hazard and accelerated failure times and (iii) Flexible parametric survival regression models with frailty generally referred to as the Royston-Parmar (RP) models and their equivalents without frailty.

## Methods and statistical models

### Cox proportional hazard models with frailty

The Cox proportional hazard (PH) model with frailty is an extension of the Cox PH model developed in 1972 which assumed that hazards are multiplicatively proportional to baseline hazards [5] as shown in Eq (1).

$$h(t) = h_0(t)e^{\beta_1 x_1 + \beta_2 x_2 + \cdots \beta_k x_k} \tag{1}$$

The above equation provides estimates of $\beta_1, \beta_2, \ldots, \beta_k$, and its variance-covariance matrix but provides no direct estimate of the baseline hazard ($h_0(t)$). However, the model provides an avenue to estimate the baseline cumulative hazard ($H_0(t)$) and baseline survival ($S_0(t)$) which can be used to estimate the $h_0(t)$ [10].

Let us assume groups i = 1,................,n groups with j = 1,................,$n_i$ observations in group $i$. For the $j^{th}$ observation in the $i^{th}$ group, the hazard is shown in Eq (2).

$$h_{ij}(t) = h_0(t)\alpha_i e^{\beta_1 x_1 + \beta_2 x_2 + \cdots \beta_k x_k} \tag{2}$$

where group-level frailty is estimated by $\alpha_i$. The frailties are unobservable positive quantities and are assumed to have a mean of 1 and a variance θ. Shared-frailty models are used to model within-group correlation; observations within a group are correlated because they share the same frailty. The degree of within-group correlation can be measured by an estimate of "$\theta$", where $\theta$ is 0 in cases where there is no frailty.

By letting $v_i = \log \alpha_i$, the hazard is as shown in Eq (3)

$$h_{ij}(t) = h_0(t)e^{\beta_1 x_1 + \beta_2 x_2 + \cdots \beta_k x_k + v_i} \tag{3}$$

which makes the log frailties $v_i$, to be analogous to random effects obtainable in the corresponding standard linear models.

Numerically, let $x_i$ be the row vector of covariates for the time interval ($t_{0i}$; $t_i$) for the $i^{th}$ observation in a dataset with N subjects (i = 1,..........,N). The estimates of the coefficient ($\beta_i$)

of the covariates ($X_i$) can be estimated by maximizing the partial log-likelihood function in Eq (4)

$$logL = \sum_{j=1}^{D}[\sum_{i\in D_j} x_i\beta - d_j\, log\{\sum_{k\in R_j} exp(x_k\beta)\}] \tag{4}$$

Where $j$ represents the index of the ordered failure times $t_{(j)}$, j = 1;. . . . . . . . .; D; such that $D_j$ is the set of $d_j$ observations that fail at $t_{(j)}$; $d_j$ is the number of failures at $t_{(j)}$; and $R_j$ is the set of observations $k$ that are at risk at time $t_{(j)}$, that is, for all $k$ such that $t_{(0k)} < t_{(j)} \leq t_{(k)}$ [27–29].

The data for Cox shared-frailty models are usually organized into G groups with the $i^{th}$ group consisting of $n_1$ observations, i = 1,. . . . . . . . .,G. The estimation of $\theta$ is done via maximum profile log-likelihood. For fixed $\theta$, estimates of $\beta$ and $v_1$,. . . . . .,$v_G$ are obtained by maximizing the parameters in Eq (5).

$$logL(\theta) = logL_{Cox}(\beta, v_1, \ldots..v_G)$$
$$+ \sum_{i=1}^{G}\left[\frac{1}{\theta}\{v_1 - e^{v_1}\} + \left(\frac{1}{\theta} + D_I\right)\left\{1 - log\left(\frac{1}{\theta} + D_I\right)\right\} - \frac{log\theta}{\theta} + log\Gamma\left(\frac{1}{\theta} + D_I\right) - log\Gamma\left(\frac{1}{\theta}\right)\right] \tag{5}$$

where $D_i$ is the number of death events in group $i$ and $logL_{Cox}(\beta, v_1, \ldots . . . . . . ,v_G)$ is the standard Cox partial log-likelihood, with the $v_i$ as the vector of the variables' coefficients indicator which identifies the groups. The $j^{th}$ observation in the $i^{th}$ group has log relative hazard $\beta x_{ij}+v_i$. The values that maximize $logL(\hat{\theta})$ are the final estimates of $\beta$ in $v_i$ [30].

## Mixed-effects parametric survival models

The mixed-effects parametric survival models otherwise called the multilevel parametric survival models are well known [31]. These models contain both fixed and random effects. The accelerated failure-time (AFT) model and the multiplicative or proportional hazards (PH) model are the most-used models for adjusting survivor functions for the effects of covariates. In the AFT model, log $t$ is expressed as a linear function of the covariates, when random-effects is incorporated, the function yields the function in Eq 6.

$$logt_{ji} = X_{ji}\beta + z_{ji}u_j + v_{ji} \tag{6}$$

for j = 1,. . . . . . . . . . . .,M, clusters, with cluster $j$ consisting of i = 1,. . . . . . . . . . . .,$n_j$ observations. The 1 X p row vector $X_{ji}$ contains the covariates for the fixed effects, with regression coefficients (fixed effects) $\beta$. The $z_{ji}$ has 1 x q dimension and contains the covariates corresponding to the random effects. Also, $v_{ji}$ are the observation-level errors with density $\varphi(.)$. In the PH models, the model contains the covariates which have a multiplicative effect on the hazard function in Eq (7).

$$h(t_{ji}) = h_0(t_{ji})exp(X_{ji}\beta + z_{ji}u_j) \tag{7}$$

where $h_0(t)$, the baseline hazard function, is assumed to be parametric. Both the exponential and Weibull models can be implemented using the AFT and PH parameterizations, but the gamma and log-logistics and log-normal can only be implemented in AFT and implemented with "mestreg" in Stata.

## Flexible parametric survival regression models

The FPSR model is based on a series of models that are modifications of several standard survival models [21] but has additional flexibility [21,23]. These models use restricted cubic splines to model a transformation of the survival function. The Weibull model is one of the

most common parametric models and is an approximation of the PH model. It has been criticized for inflexibility in the shape of the baseline hazard function, which either increases or decreases monotonically. Weibull survival function is $S(t) = \exp(-\lambda t\gamma)$, and the corresponding log cumulative hazard scale is $\ln\{H(t)\} = \ln[-\ln\{S(t)\}] = \ln[-\ln\{\exp(-\lambda t\gamma)\}] = \ln(\lambda) + \gamma\ln(t)$, after transformation. Addition of covariates to the model produces $\ln\{H(t|x_i)\} = \ln(\lambda) + \gamma\ln(t) + x_i\beta$.

Splines are flexible mathematical functions defined by piecewise polynomials with some constraints to ensure that the overall curve is smooth, are used. The polynomials join one another at points called knots. The fitted function is forced to have continuous $0^{th}$, $1^{st}$ and $2^{nd}$ derivatives. The most common splines used are cubic splines. Restricted cubic splines with k knots can be fitted by creating *k-1* derived variables. For knots $k_1, k_2, \ldots\ldots\ldots\ldots..k_k$, and parameters $\gamma_0, \gamma_1, \ldots\ldots\ldots\ldots\ldots.\gamma_{k-1}$, a restricted cubic spline function can be written as $s(x) = \gamma_0 + \gamma_1 z_1 + \cdots\ldots\ldots\ldots\ldots.+\gamma_{k-1}z_{k-1}$; where $z_1 = x = \ln(t)$ and $z_j (j\geq 2)$. The derived variables, $z_j$, are computed as in Eq (8)

$$z_j = (x - k_j)_+^3 - \phi_j(x - k_j)_+^3 - (1 - \phi_j)(x - k_j)_+^3; \tag{8}$$

where $j = 2, \ldots, k-1$; $(x - k_j)_+^3 = max\{0, (x - a)^3\}$; $\phi_j = (k_k - k_j)/(k_k - k_1)$; $k_k$ is the maximum k, and $k_1$ is the minimum k. The derived variables can be highly correlated and are orthogonalized by using Gram–Schmidt orthogonalization.

The hazard function involves the derivatives of the restricted cubic splines functions as $s'(x) = \gamma_1 z'_1 + \gamma_2 z'_2 + \cdots\ldots\ldots\ldots.+\gamma_{k-1}z'_{k-1}$. The choice of position of knots determines the complexity of the flexible models. Usually, *k* knots, maximum at 9 knots, has *k+1* degrees of freedom (df). The position of the knots (internal) is usually in centiles computed as 100/df. So, for 3 knots, the df is 4 and the knots will be located at centiles 25, 50 and 75. The internal knots are bounded by "boundary knots" which are placed at the minimum and maximum of the distribution of uncensored survival times. The FPSR models become the Weibull model if the number of knots is 0, while $\gamma_0$ and $\gamma_1$ are equal to the scale parameter and shape parameter respectively. Royston et al. suggest using 1 or 2 knots for smaller (<10,000) datasets and 4 or 5 for larger (> = 10,000) datasets [10]. The FSPR models are implemented in Stata using "stpm2" with an option for frailty.

In the flexible parametric model, the contribution to the log-likelihood for the $i^{th}$ individual on the log cumulative hazard scale can be written as shown in Eq 9.

$$\ln L_i = d_i(ln[s'\{\ln(t_i)|\gamma, k_0\}] + \eta_i) - \exp(\eta_i) \tag{9}$$

where $d_i$ is the event indicator. The likelihood can be maximized by defining an additional equation for the derivatives of the spline function and constrain the parameters to be equal to the equivalent spline functions in the main linear prediction [22,32].

## Model selection criteria

Log-likelihood, Akaike information criteria (AIC) [33] and the Bayesian information criteria (BIC) [34] were used for model selection. Lower values of AIC and BIC indicated a better model fit. AIC and BIC are usually computed and compared separately among different models to determine the best fitting model. However, confusion may arise if the best fitting model according to the AIC is different from that identified by the BIC [14]. Literature suggests that AIC will choose a more complex model irrespective of sample size while BIC is more likely to choose a simpler model [14]. AIC is often preferable in situations when a false negative finding would be considered to be more misleading than a false positive, and BIC is superior in situations where a false positive is as misleading as, or more misleading than, a false negative. AIC

is best for prediction as it is asymptotically equivalent to cross-validation. BIC is best for an explanation as it allows consistent estimation of the underlying data generating process [14,35].

## The data

The dataset used for this study originates from a project evaluating the effectiveness of dental implant therapy in Sweden. A cohort of 596 randomly selected adults, provided with implant-supported dental restorations in 2003, were followed over 9 years. The extent of dental treatment varied from the replacement of single teeth to the restoration of full jaws. The average number of implants per patient was 4.0 ±2.8 (range 1–12). Complications related to the restorations/implants were scored using the patient as the unit of analysis and timing was recorded in days relative to the time point of implant insertion. The complications included: Loss of a dental implant, development of peri-implantitis and/or occurrence of a technical complication. For details regarding case definitions of the different categories of complications, the reader is referred to Derks et al. [25,26,36]. The occurrence of any of the complications referred to above was considered as an event in the present analyses. There were a total of 1,038 events during the observation period with 469 complications in single-record/single-failure data.

## Operational definitions

Median survival time: This is a statistic that refers to how long patients "survive" in general after dental restorative therapy including the use of implants.

The incidence rate is a measure of the frequency with which dental implant complications occurred per day.

$$Incidence\ Rate = \frac{\text{Number of new cases of disease during specified period}}{\text{Time each person was observed, totalled for all persons}}$$

## Ethics approval and patient consent

The research protocol was approved by the regional Ethical Committee, Gothenburg, Sweden (Dnr 290–10), registered at ClinicalTrials.gov (NCT01825772) and study participants signed an informed consent form prior to inclusion.

## Results

One of the 596 subjects was excluded from analysis due to missing data. The mean age (in 2003) of the 595 included participants was 62.3 (SD = 9.3) years, with 42% aged 60–69 years, 24% aged 70–79 years and 55% were females. Roughly 60% of patients presented without signs of periodontitis at the 9-year examination, 24% had periodontitis and 16% were edentulous (no teeth). A total of 28% had full-jaw restorations, 48% had partial-jaw restorations and 24% had single crowns, only. Regarding dental products, 31% received Straumann implants (Type A), 20% had Astra Tech implants (Type B), 40% had Nobel Biocare implants (Type C). The remaining 9% of subjects were treated with various other types of implants categorized as Type D.

The overall incidence rate of implant complications was 0.000241 per day. It was higher among those without remaining natural teeth (0.000387) and those who had full-jaw restorations (0.000439). The median survival time (when 50% of implant-carrying subjects would have "failed") to implant complications was 2476 days, while the 25% survivorship was 820 days. The medium "survival" time was highest among those who had partial-jaw restorations

(3347 days), females (3044 days), treated within the public dental service (3044 days), treated with Type A (3347 days) or other dental products (Type D) (3227 days respectively) as shown in Table 1.

The distribution of the incidence rate of implant complications by time is shown in Fig 1. The incidence rate reduced with increasing time of follow-up.

## The hazard and survival functions of dental complications under different distributions

We assessed the hazard function under different models for periodontal status (one of the main prognostic factors) in Fig 2. In Fig 2, the chart in panel (a) is from the Weibull mixed effects parametric proportional hazard (b) Loglogistic mixed effects parametric proportional hazard (c) Cox proportional hazard and (d) Cox smoothed proportional hazard. The hazard for "periodontitis" was consistently higher than the hazard for "healthy" and "no teeth". The hazard for "healthy" and "no teeth" was similar.

In Fig 3, we present the survival function under different models for periodontal status. The chart in panel (a) is from the Weibull mixed effects parametric proportional hazard (b) Loglogistic mixed effects parametric proportional hazard (c) Cox proportional hazard and (d) Cox smoothed proportional hazard for periodontal status. The survival for "periodontitis" was consistently lower than the survival for "healthy" and "no teeth". The survival for "healthy" and "no teeth" was similar.

## Test of assumption of proportionality

The test of violations of assumptions of the proportional hazard showed that the test was not violated ($X^2$ = 5.50, df = 4, p = 0.240)

## Comparison of the survival and hazard functions of the flexible model at different degrees of freedom

We compared the performance of the survival and hazard functions of the flexible model at various degrees of freedom (1, 2, 3 and 6) for the periodontal status of the patients. A degree of freedom of 1 is an equivalent of the Weibull distribution. The hazard functions of the Weibull distribution were different from the hazard functions at the other degrees of freedom. However, the function at 6 degrees of freedom was different and more flexible than at 2 and 3 degrees of freedom (Fig 4A). The survival functions at 2, 3 and 6 degrees of freedom were, however, similar but distinct at 1 degree of freedom (Fig 4B).

## Selection of the best model

Loglikelihood, AIC BIC for all the models considered, with and without frailty, are presented in Table 2. All three parameters were consistently lower among the flexible frailty models at different degrees of freedom than the Cox PH frailty, parametric frailty models (Table 2). We observed that the AIC and BIC of the parametric models without frailty were consistently lower than those with frailty. Among the FPSR models at different degrees of freedom, the lowest loglikelihood was at df = 6, the lowest AIC was at df = 4, while the lowest BIC was at df = 2. However, for df >1, differences between the lowest and highest loglikelihood, between the lowest and highest AIC and between the lowest and highest BIC were 4.4 (0.45%), 1.47 (0.04%) and 18.7 (0.89%) respectively. According to the AICs, all the FPSR models at df>2 were similar. Hence we chose the simplest of all the FPSR models at df = 2. Our decision was further supported by the significance of the spline variables for the log baseline cumulative hazard

**Table 1. Distribution of incidence rate and quartile survival times by patients' characteristics (n = 595).**

| Characteristics | N(%) | Days at risk | incidence rate | Survival time (days) | | |
|---|---|---|---|---|---|---|
| | | | | 25% | Median (50%) | 75% |
| Age(years) in 2003 mean(sd) | 62.3(9.3) | | | | | |
| <50 | 80(13.5) | 266436 | 0.000135 | 1935 | cbc | cbc |
| 50–59 | 121(20.3) | 399995 | 0.000170 | 1185 | cbc | cbc |
| 60–69 | 252(42.4) | 813359 | 0.000299 | 746 | 1874 | cbc |
| 70–79 | 145(24.4) | 467601 | 0.000261 | 593 | 1927 | cbc |
| Age group | | | | | | |
| Younger (<60) | 201(33.8) | 666431 | 0.000156 | 1386 | cbc | cbc |
| Older (≥60) | 394(66.2) | 1280960 | 0.000285 | 685 | 1915 | cbc |
| Sex | | | | | | |
| Male | 267(44.9) | 872344 | 0.000288 | 685 | 1927 | cbc |
| Female | 328(55.1) | 1075047 | 0.000203 | 899 | 3378 | cbc |
| Periodontal status | | | | | | |
| Healthy | 356(59.8) | 1168495 | 0.000187 | 1081 | cbc | cbc |
| Periodontitis | 144(24.2) | 468620 | 0.000280 | 868 | 1966 | cbc |
| No Teeth | 95(16.0) | 310276 | 0.000387 | 471 | 1185 | cbc |
| Extent of treatment | | | | | | |
| Full jaw | 167(28.1) | 537496 | 0.000439 | 470 | 1082 | 2840 |
| Partial jaw | 283(47.6) | 928468 | 0.000208 | 929 | 3347 | cbc |
| Single | 145(24.4) | 481427 | 0.000083 | . | cbc | cbc |
| Smoker | | | | | | |
| Yes | 76(12.8) | 246840 | 0.000292 | 929 | 2118 | cbc |
| No | 519(87.2) | 1700551 | 0.000234 | 820 | 2509 | cbc |
| Clinical setting | | | | | | |
| Public dental service | 174(29.2) | 575246 | 0.000214 | 868 | 3044 | cbc |
| Private dental service | 372(62.5) | 1211970 | 0.000261 | 807 | 2210 | cbc |
| Mix | 49(8.2) | 160175 | 0.000187 | 594 | cbc | cbc |
| Frequency of maintenance | | | | | | |
| Regular (annual) | 479(82.2) | 1565319 | 0.000259 | 746 | 2057 | cbc |
| Irregular (< annual) | 104(17.8) | 341940 | 0.000178 | 1752 | cbc | cbc |
| Ever Smoker | | | | | | |
| Yes | 209(35.1) | 682082 | 0.000238 | 746 | 2515 | cbc |
| No | 386(64.9) | 1265309 | 0.000243 | 820 | 2280 | cbc |
| Dental product | | | | | | |
| Type A | 181(31.2) | 605090 | 0.000207 | 959 | 3347 | cbc |
| Type B | 115(19.8) | 374929 | 0.000272 | 746 | 1874 | cbc |
| Type C | 230(39.6) | 740492 | 0.000236 | 929 | 2604 | cbc |
| Typ D | 55(9.5) | 180580 | 0.000222 | 869 | 3227 | |
| Bone augmentation | | | | | | |
| No | 436(84.7) | 1428524 | 0.000240 | 820 | 2476 | cbc |
| Yes | 79(15.3) | 256548 | 0.000265 | 654 | 2070 | cbc |
| Retention of restoration | | | | | | |
| Cemented | 198(34.1) | 664274 | 0.000117 | 2057 | cbc | cbc |
| Screw-retained | 346(59.6) | 1122931 | 0.000314 | 624 | 1661 | cbc |
| Both | 37(6.3) | 119748 | 0.000275 | 404 | 2069 | cbc |

(*Continued*)

**Table 1.** (Continued)

| Characteristics | N(%) | Days at risk | incidence rate | Survival time (days) | | |
|---|---|---|---|---|---|---|
| | | | | **25%** | **Median (50%)** | **75%** |
| Total | 595 | 1947391 | 0.000241 | 820 | 2476 | cbc |

cbc Cannot be computed

Missing data if the sum of categories <595.

(_rcsi) otherwise called the slope of the hazard curve within each of the knots generated by the degrees of freedom. At df = 2, the two slopes were statistically significant: _rcs1 = 2.26 (p<0.001) and _rcs2 = 1.150 (p<0.001). At df = 3, the first two slopes were statistically significant: _rcs1 = 2.25 (p<0.001) and _rcs2 = 1.134 (p<0.001) but the last slope was not significant (_rcs3 = 1.02 (p = 0.107). The slopes at df>2 had similar patterns with the slopes at df = 3.

## Modelling the risk factors of implant complications

We fitted an FPSR model at 2 degrees of freedom to the data and identified the adjusted determinants of implant complications among the patients. Table 3 showed that the adjusted hazard of implant complications was 45% (adjusted Hazard Ratio (aHR) = 1.449; 95% Confidence Interval (CI): 1.153–1.821, p = 0.001) higher among patients with periodontitis than those who were periodontally healthy. While controlling for other variables, the hazard of implant complications was about 5 times (aHR = 4.641; 95% CI: 2.911–7.401, p<0.001) and 2 times (aHR = 2.338; 95% CI: 1.553–3.519, p<0.001) higher among patients with full- and partial-jaw restorations respectively when compared to subjects with single crowns, only. The adjusted hazard of implant complications was 27% (aHR = 1.272; 95% CI: 1.047–1.548, p = 0.016) higher among male patients than females. The adjusted hazard of an implant complication was 40% (aHR = 1.397; 95% CI: 1.069–1.826, p = 0.014) higher among patients provided with Type B dental products with Type A products as reference. Smoking history, type of retention and age were not significant predictors of complications.

## Discussion

This study was designed to apply and compare the performance of semi-parametric, parametric and flexible parametric survival regression models to a dataset on dental implant-related

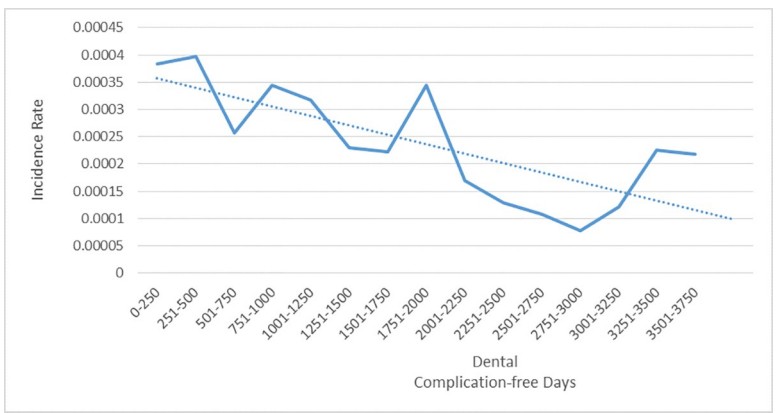

**Fig 1. Distribution of incidence rate of dental failure by time.**

### Weibull PH hazard for Periodontal status

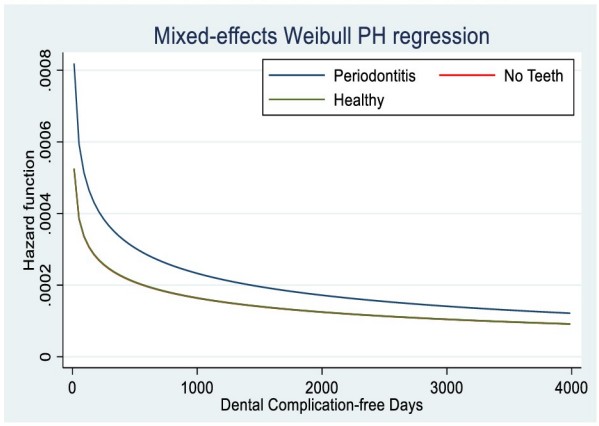

### Loglogistic hazard for Periodontal status

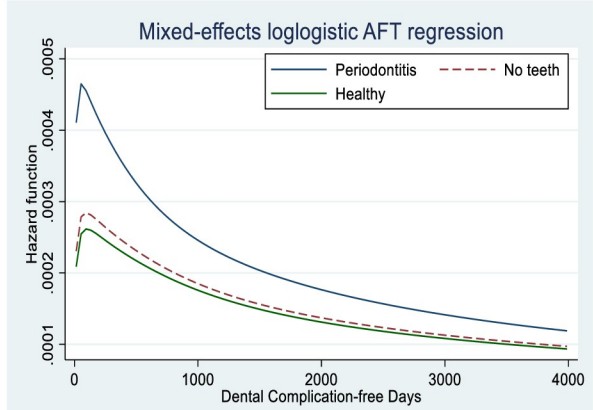

### Cox hazard no str for Periodontal status

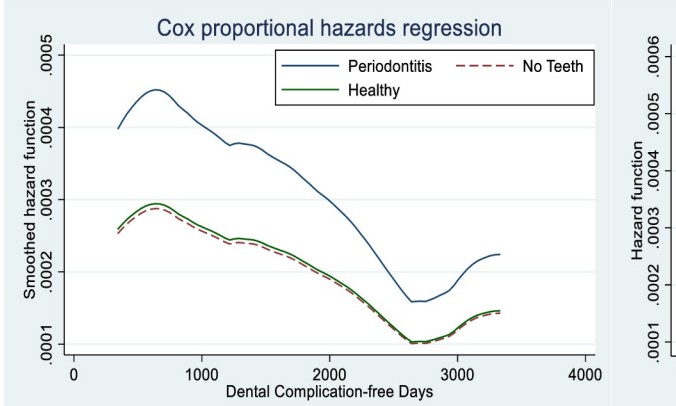

### Cox smoothed hazard for Periodontal status

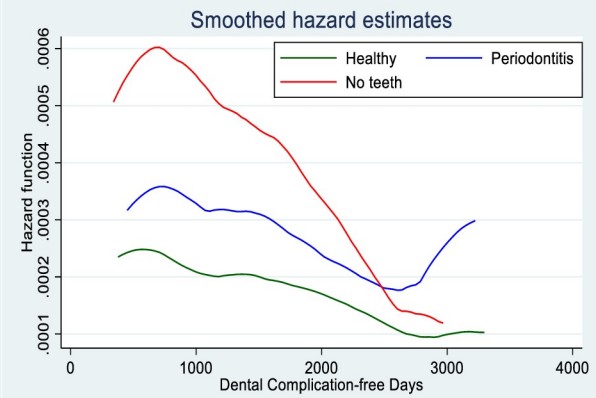

**Fig 2. Comparison of the hazard functions of the models using the determinate variable.**

complications with or without frailty. This analytical study showed that models with frailty performed better than those without frailty except among the parametric models where the reverse was the case. This could be ascribed to inconsistencies and inflexibilities of parametric models (Royston and Lambert, 2011). Nonetheless, the AIC and the BIC of the flexible models were lower than those computed from the other models irrespective of whether the clustering nature of the implant data was considered or not.

Therefore, the flexible parametric survival regression model was the best of the three main models considered in this study. Our finding is consistent with findings in earlier studies [10,15,23]. All measures of model fit and model selection adopted in our study were consistently better in the flexible parametric survival regression models than in the other models. Loglikelihood, AIC and BIC were lower in the flexible parametric survival regression models than the Cox PH model and the parametric models. Similar findings have been reported in the literature [15,23,37]. The flexible models had a unique advantage by separating the hazard function into segments (splines) based on the specified degrees of freedom and computing the hazard within each spline [9,10,18,21].

We used AIC and BIC to select the ultimate degrees of freedom to use for the flexible parametric survival regression model. AIC and BIC are measures of the amount of information lost in the models [33,34]. The lower these values, the better the models. In this study, BIC

a. Weibull PH Survival function for Periodontal status

b. Loglogistic  Survival function for Periodontal status

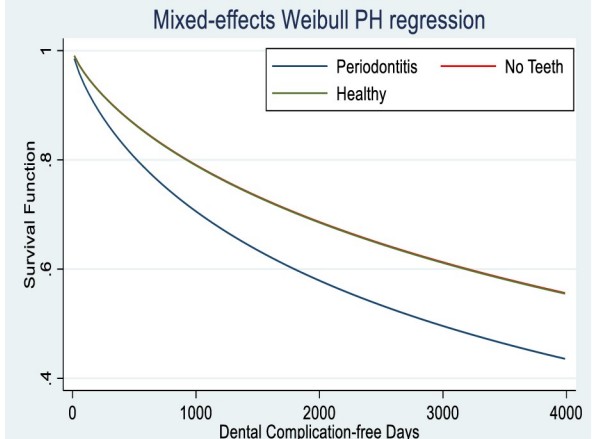

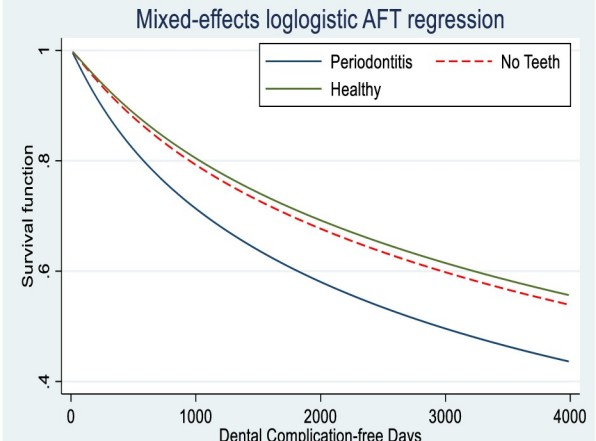

c. Cox Survival function for Periodontal status

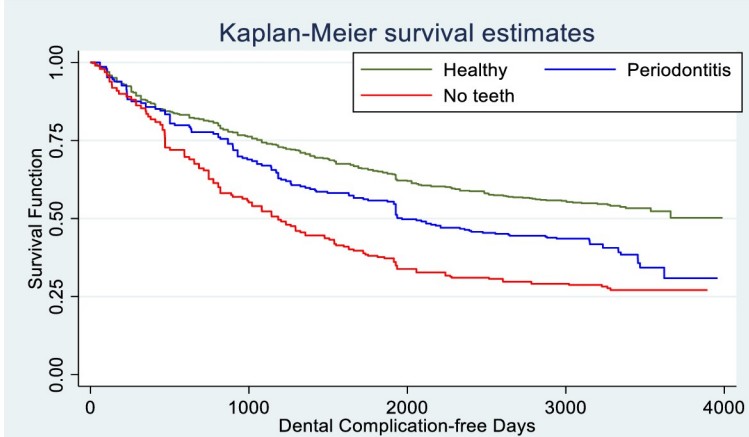

**Fig 3. Comparison of the survival functions of the models using the determinate variable.**

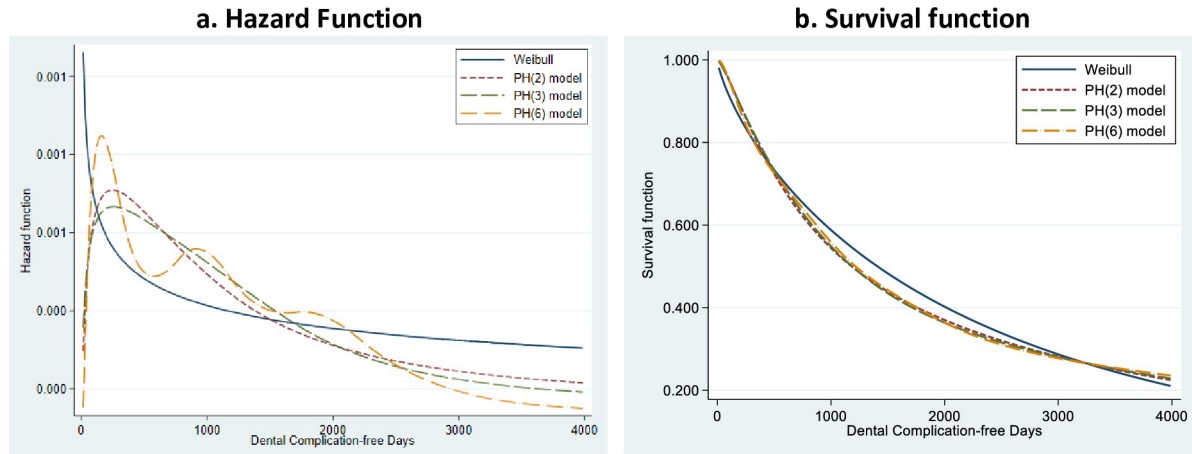

**Fig 4. Comparison of the survival and hazard functions of the Weibull model and the flexible models.**

**Table 2. Comparison of the flexible models using different knots.**

| Frailty | Model | Model | ll(null) | ll(model) | df | AIC | BIC |
|---|---|---|---|---|---|---|---|
| None | Cox | Cox | -2717.70 | -2633.63 | 17 | 5301.26 | 5384.09 |
| | Parametric | Weibull | -1187.65 | -1134.54 | 19 | 2307.07 | 2399.64 |
| | | Exponential | -1189.12 | -1134.86 | 18 | 2305.73 | 2393.43 |
| | | Log Logistic | -1182.18 | -1130.25 | 19 | 2298.51 | 2391.08 |
| | | Gamma | -1178.59 | -1130.46 | 20 | 2300.92 | 2398.37 |
| | Flexible Model | df = 1 | -1294.47 | -1134.54 | 19 | 2307.07 | 2399.64 |
| | | df = 2 | -1294.47 | -1128.29 | 20 | 2296.58 | 2394.03 |
| | | df = 3 | -1294.47 | -1127.95 | 21 | 2297.90 | 2400.21 |
| | | df = 4 | -1294.47 | -1125.68 | 22 | 2295.36 | 2402.55 |
| | | df = 5 | -1294.47 | -1124.84 | 23 | 2295.69 | 2407.74 |
| | | df = 6 | -1294.47 | -1123.84 | 24 | 2295.68 | 2412.62 |
| Yes | Cox | Cox with strata | -2047.18 | -2001.85 | 17 | 4037.70 | 4120.52 |
| | | Cox frailty | -2866.48 | -2577.24 | 17 | 5188.48 | 5271.31 |
| | Parametric | Weibull | -4359.71 | -3900.89 | 20 | 7841.78 | 7939.22 |
| | | Exponential | -4376.43 | -3910.82 | 19 | 7859.64 | 7952.21 |
| | | Log Logistic | -4349.04 | -3898.90 | 20 | 7837.80 | 7935.24 |
| | | Gamma | -4363.40 | -3905.27 | 20 | 7850.54 | 7947.98 |
| | Flexible Model | df = 1 | -1143.70 | -988.24 | 19 | 2014.47 | 2107.04 |
| | | df = 2 | -1143.70 | -972.51 | 20 | 1985.01 | 2082.45 |
| | | df = 3 | -1143.70 | -971.73 | 21 | 1985.45 | 2087.76 |
| | | df = 4 | -1143.70 | -969.99 | 22 | 1983.98 | 2091.17 |
| | | df = 5 | -1143.70 | -969.16 | 23 | 1984.31 | 2096.37 |
| | | df = 6 | -1143.70 | -968.11 | 24 | 1984.22 | 2101.15 |

AIC Akaike Information Criteria BIC Bayesian Information Criteria df degrees of freedom ll loglikelihood PH Proportional Hazard AFT Accelerated Failure Rate

was lowest at 2 degrees of freedom while AIC was lowest at 4 degrees of freedom. This discrepancy has previously been reported [35] and may be due to how the two information measures compute "complexities". The problem of defining "N" (the number of observations) is not related to AIC because N is not used in computing AIC, which rather uses a constant 2 to weight complexity as measured by k (number of parameters estimated), rather than ln(N) in BIC. According to Stone at al. [38], AIC approximately minimizes the prediction error and is asymptotically equivalent to leave-1-out cross-validation (LOOCV) while BIC is equivalent to leave-k-out cross-validation (LKOCV) [39] and it is not consistent with the amount of data available. However, BIC has the advantage of being consistent. With a very large amount of data, and if the true model is among the candidate models, the probability of selecting the true model based on the BIC criterion would approach 1. This, however, may slightly affect prediction performance. We chose the flexible parametric survival regression model at 2 degrees of freedom as suggested by the BIC because the slopes of the curves within each spline were insignificant after 2 degrees of freedom, the differences in parameter estimates at 2, 3 and 4 degrees of freedom were negligible. Also, the AIC at 2, 3 and 4 degrees of freedom changed by 0.07%, which was considered negligible.

Our finding that the FPSR method with frailty fitted the data used in this study is further corroborated by the behaviour of the hazard and survival functions shown in Figs 2–4. However, there could be challenges of over-parametrization in the flexible model due to its adaptability and incorporation of up to ten knots. Also, the Cox model makes minimal assumptions about the form of the baseline hazard function and may have hindered the prediction of

**Table 3. Adjusted prognostic factors of dental implant complications using FPSR model (df = 2).**

| Characteristics | Adjusted Hazard Ratio | 95% CI | p-value |
|---|---|---|---|
| Periodontal status | | | |
| Healthy | 1.000 | | |
| Periodontitis | 1.449 | 1.153–1.821 | 0.001 |
| No teeth | 1.050 | 0.797–1.383 | 0.730 |
| Extent of treatment | | | |
| Full jaw | 4.641 | 2.911–7.401 | <0.001 |
| Partial jaw | 2.338 | 1.553–3.519 | <0.001 |
| Single | 1.000 | | |
| Age (years) in 2003 | | | |
| <50 | 1.000 | | |
| 50–59 | 1.086 | 0.713–1.652 | 0.702 |
| 60–69 | 1.108 | 0.740–1.658 | 0.620 |
| 70–79 | 0.860 | 0.559–1.322 | 0.491 |
| Gender | | | |
| Male | 1.272 | 1.047–1.548 | 0.016 |
| Ever smoker | | | |
| Yes | 1.014 | 0.769–1.337 | 0.922 |
| Dental product | | | |
| Type A | 1.000 | | |
| Type B | 1.397 | 1.069–1.826 | 0.014 |
| Type C | 1.074 | 0.848–1.360 | 0.554 |
| Type D | 1.116 | 0.779–1.597 | 0.550 |
| Retention of restoration | | | |
| Screw-retained | 1.000 | | |
| Cemented | 0.870 | 0.627–1.208 | 0.406 |
| Both | 0.920 | 0.607–1.395 | 0.695 |
| _rcs1 | 2.262 | 2.080–2.459 | <0.001 |
| _rcs2 | 1.150 | 1.091–1.213 | <0.001 |

_rcs are the spline variables for the log baseline cumulative hazard

hazards and other related functions for a given set of covariates. It also results in unsmooth estimated curves and lack of information about what occurs between the observed failure times. Parametric models, on the other hand, produce smooth predictions by assuming a functional form of the hazard. Its assumed form is too structured for use with real data (Royston and Lambert, 2011). Therefore, the non-proportional hazards can be modelled using restricted cubic splines in FPSR models [14] and thereby produce a better fit.

The fitted flexible parametric survival regression model at 2 degrees of freedom showed that the hazard of implant complications was higher among male patients, patients with periodontitis, among patients with either full- or partial-jaw restorations and among patients that were provided with dental product Type B. It is plausible that more extensive restorations are at higher risk for complications through the simple fact that more implants and surfaces are exposed to potential events. More extensive restorations, however, may also serve as a surrogate parameter for the individual's susceptibility to developing tooth- or implant-related problems. This may be illustrated by the fact that subjects presenting with periodontitis at remaining teeth were at higher risk for implant-related complications. This relationship is most likely explained by the strong association between periodontitis and peri-implantitis

[40]. Peri-implantitis was one of the complications recorded in the present study. The background to the other factors identified in the model (sex and dental product) are not understood. It may be speculated that biting force and/or behaviour in terms of oral health may have had an impact.

The covariates included in the flexible model have shown that there is a wide range of factors that contribute to complications affecting dental implants. Their inclusion has influenced the performance of the models as they demonstrated reality. For instance, the risk of implant complications was generally higher among patients with periodontitis than those that were periodontally healthy. No difference, however, was noted between periodontally healthy and edentulous patients. Similar assertions have been made in earlier studies [41,42].

## Conclusion

Flexible parametric survival model represents the best approach for estimating the hazard of clustered implant complications including (i) implant loss, (ii) peri-implantitis and (iii) technical complications. The study underscores the need to explore the multilevel (clustering) nature of datasets to be analysed. Non-consideration of the clustering nature of data is potentially misleading. The hazard of complications was higher among male patients, patients with periodontitis, patients with more extensive restorations and was dental product specific.

## Acknowledgments

The authors appreciate the logistic supports provided by the Consortium for Advanced Research and Training in Africa (CARTA) to AFF to visit the University of Gothenburg as part of his fellowship at the University of Warwick.

## Author Contributions

**Conceptualization:** Adeniyi Francis Fagbamigbe, Karolina Karlsson, Max Petzold.

**Data curation:** Adeniyi Francis Fagbamigbe, Max Petzold.

**Formal analysis:** Adeniyi Francis Fagbamigbe, Max Petzold.

**Investigation:** Adeniyi Francis Fagbamigbe, Karolina Karlsson, Jan Derks.

**Methodology:** Adeniyi Francis Fagbamigbe.

**Supervision:** Jan Derks, Max Petzold.

**Visualization:** Adeniyi Francis Fagbamigbe.

**Writing – original draft:** Adeniyi Francis Fagbamigbe, Karolina Karlsson, Jan Derks, Max Petzold.

**Writing – review & editing:** Adeniyi Francis Fagbamigbe, Karolina Karlsson, Jan Derks, Max Petzold.

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
