## [Decision Letter · Decision Letter 0]

27 Nov 2020

PONE-D-20-32590

Performance Evaluation of survival regression models in analysing Swedish dental implant complication data with frailty

PLOS ONE

Dear Dr. Fagbamigbe,

Thank you for submitting your manuscript to PLOS ONE. After careful consideration, we feel that it has merit but does not fully meet PLOS ONE’s publication criteria as it currently stands. Therefore, we invite you to submit a revised version of the manuscript that addresses the points raised during the review process.

We look forward to receiving your revised manuscript.

Kind regards,

Feng Chen

Academic Editor

PLOS ONE

Journal Requirements:

2. Please correct your reference of  "p=0.000" to "p<0.001" or as similarly appropriate, as p values cannot equal zero.

5. We note you have included a table to which you do not refer in the text of your manuscript. Please ensure that you refer to Table 3 in your text; if accepted, production will need this reference to link the reader to the Table.

Reviewers' comments:

Reviewer's Responses to Questions

**Comments to the Author**

1. Is the manuscript technically sound, and do the data support the conclusions?

Reviewer #1: Yes

Reviewer #2: Yes

Reviewer #3: Yes

2. Has the statistical analysis been performed appropriately and rigorously? 

Reviewer #1: Yes

Reviewer #2: Yes

Reviewer #3: Yes

3. Have the authors made all data underlying the findings in their manuscript fully available?

Reviewer #1: Yes

Reviewer #2: Yes

Reviewer #3: No

4. Is the manuscript presented in an intelligible fashion and written in standard English?

Reviewer #1: Yes

Reviewer #2: Yes

Reviewer #3: Yes

5. Review Comments to the Author

Reviewer #1: This paper evaluated the performance of 13 survival regression models in analyzing dental implant complication based on a case study in Sweden. The topic is interesting and worth of investigation. The results indicate that the flexible parametric survival model with frailty outperforms the other alternatives. The findings provide a good suggestion on method selection for modeling dental implant complication. Overall, the paper is generally well organized and qualified to be published in PLOS ONE.

Reviewer #2: This paper evaluated the performance of 13 survival regression models in assessing the factors associated with the timing of complications in implant-supported dental restorations in a Swedish cohort. The topic is interesting. The methods sound. The results are meaningful and useful. One only issue is that the resolution of some figures could be increased.

Reviewer #3: This paper is written in a way that takes the reader through the models very well.

I have some minor comments:

I am not familiar with dental data, so I didn't have a good idea about what an event was, and what a complication was in the sentence of: line 223: "There were a total of 1,038 events during the observation period with 469 complications in single-record/single-failure data".

line 225-230: Is the model improvement assessed in line with how complicated the model is? As in, typically in multivariate models a new term is only included if the improvement in model fit is significant, as a particular level of significance. Was this considered here? It may be worth stating that the degree of improvement is not relevant in selecting a best fitting model. Reading the results section it does seem like there was a consideration for how complicated the model was.

line 285: is it relevant to have the p-value to 4 decimal places? Wouldn't 2 do?

Table 3: are the estimates too precise? Would 2 decimal places for the hazard ratio and confidence intervals still provide the same relevant information, and p-values to 2 significant figures? (or 1 if p<0.01)

Also, present p=0.000 as p<0.001.

6. PLOS authors have the option to publish the peer review history of their article (what does this mean?). If published, this will include your full peer review and any attached files.

Reviewer #1: No

Reviewer #2: No

Reviewer #3: No

---

## [Author Response · Author response to Decision Letter 0]

8 Dec 2020

November 28th 2020

Dear Editor

PLOS ONE 

Through 

Feng Chen

Academic Editor 

PONE-D-20-32590

Performance Evaluation of survival regression models in analysing Swedish dental implant complication data with frailty

We the authors of above mentioned paper appreciate the efforts and comments of the editor and the eminent reviewers. We have addressed all these comments. A point-by-point response to the issues in our revised manuscript is listed below.

Please note that we used the file with the tracked changes to describe where the changes were made. 

Editorial Comments

 Thank you. We have reformatted the manuscript in line with Plos one requirements

2. Please correct your reference of "p=0.000" to "p<0.001" or as similarly appropriate, as p values cannot equal zero.

 Thank you we have changed all its appearance

This is human research based on sensitive data which cannot be publicly shared. Data contain potentially identifying and sensitive patient information. The interested party is referred to the regional Ethical Committee, Gothenburg, Sweden (PO Box 401, 405 30 Gothenburg, Sweden) and/or the University of Gothenburg, Gothenburg, Sweden (PO Box 100, 405 30 Gothenburg, Sweden). 

Kindly see the comment above.

 Thank you. It has been moved

5. We note you have included a table to which you do not refer in the text of your manuscript. Please ensure that you refer to Table 3 in your text; if accepted, production will need this reference to link the reader to the Table.

 Thank you. It has been moved

Reviewers' comments:

Reviewer's Responses to Questions

Comments to the Author

1. Is the manuscript technically sound, and do the data support the conclusions?

Reviewer #1: Yes

Reviewer #2: Yes

Reviewer #3: Yes

Thank you all.

2. Has the statistical analysis been performed appropriately and rigorously?

Reviewer #1: Yes

Reviewer #2: Yes

Reviewer #3: Yes

Thank you all.

3. Have the authors made all data underlying the findings in their manuscript fully available?

Reviewer #1: Yes

Reviewer #2: Yes

Reviewer #3: No

Thank you all.

4. Is the manuscript presented in an intelligible fashion and written in standard English?

Reviewer #1: Yes

Reviewer #2: Yes

Reviewer #3: Yes

Thank you all.

5. Review Comments to the Author

Reviewer #1: This paper evaluated the performance of 13 survival regression models in analyzing dental implant complication based on a case study in Sweden. The topic is interesting and worth of investigation. The results indicate that the flexible parametric survival model with frailty outperforms the other alternatives. The findings provide a good suggestion on method selection for modeling dental implant complication. Overall, the paper is generally well organized and qualified to be published in PLOS ONE.

Thank you all.

Reviewer #2: This paper evaluated the performance of 13 survival regression models in assessing the factors associated with the timing of complications in implant-supported dental restorations in a Swedish cohort. The topic is interesting. The methods sound. The results are meaningful and useful. One only issue is that the resolution of some figures could be increased.

Reviewer #3: This paper is written in a way that takes the reader through the models very well.

Thank you all.

I have some minor comments:

I am not familiar with dental data, so I didn't have a good idea about what an event was, and what a complication was in the sentence of: line 223: "There were a total of 1,038 events during the observation period with 469 complications in single-record/single-failure data".

The complications considered in the present analysis included (i) loss of an implant, (ii) development of peri-implantitis and (iii) technical problems (lines 220-222). The occurrence of any of these were considered as an “event”. We added a sentence to the relevant paragraph (lines 223-224) for clarification.

line 225-230: Is the model improvement assessed in line with how complicated the model is? As in, typically in multivariate models a new term is only included if the improvement in model fit is significant, as a particular level of significance. Was this considered here? It may be worth stating that the degree of improvement is not relevant in selecting a best fitting model. Reading the results section it does seem like there was a consideration for how complicated the model was.

Thank you. We didn’t set for model complication. We set out to select the best model irrespective of its simplicity or complexity in as much it considered frailty.

line 285: is it relevant to have the p-value to 4 decimal places? Wouldn't 2 do?

Thank you, three decimal places are usually recommended by most journals including Plos One.

Table 3: are the estimates too precise? Would 2 decimal places for the hazard ratio and confidence intervals still provide the same relevant information, and p-values to 2 significant figures? (or 1 if p<0.01)

Thank you, yes, they will provide same information. Again, the journals requirement is 3 decimal places.

Also, present p=0.000 as p<0.001.

 Thank you we have changed all its appearance

6. PLOS authors have the option to publish the peer review history of their article (what does this mean?). If published, this will include your full peer review and any attached files.

Do you want your identity to be public for this peer review? For information about this choice, including consent withdrawal, please see our Privacy Policy.

Reviewer #1: No

Reviewer #2: No

Reviewer #3: No

Thank you all.

Adeniyi Fagbamigbe

On behalf of authors

---

## [Decision Letter · Decision Letter 1]

23 Dec 2020

Performance Evaluation of survival regression models in analysing Swedish dental implant complication data with frailty

PONE-D-20-32590R1

Dear Dr. Fagbamigbe,

We’re pleased to inform you that your manuscript has been judged scientifically suitable for publication and will be formally accepted for publication once it meets all outstanding technical requirements.

Kind regards,

Feng Chen

Academic Editor

PLOS ONE

Additional Editor Comments (optional):

Reviewers' comments:

Reviewer's Responses to Questions

**Comments to the Author**

1. If the authors have adequately addressed your comments raised in a previous round of review and you feel that this manuscript is now acceptable for publication, you may indicate that here to bypass the “Comments to the Author” section, enter your conflict of interest statement in the “Confidential to Editor” section, and submit your "Accept" recommendation.

Reviewer #1: All comments have been addressed

Reviewer #2: All comments have been addressed

Reviewer #3: All comments have been addressed

2. Is the manuscript technically sound, and do the data support the conclusions?

Reviewer #1: (No Response)

Reviewer #2: (No Response)

Reviewer #3: (No Response)

3. Has the statistical analysis been performed appropriately and rigorously? 

Reviewer #1: (No Response)

Reviewer #2: (No Response)

Reviewer #3: (No Response)

4. Have the authors made all data underlying the findings in their manuscript fully available?

Reviewer #1: (No Response)

Reviewer #2: (No Response)

Reviewer #3: (No Response)

5. Is the manuscript presented in an intelligible fashion and written in standard English?

Reviewer #1: (No Response)

Reviewer #2: (No Response)

Reviewer #3: (No Response)

6. Review Comments to the Author

Reviewer #1: (No Response)

Reviewer #2: (No Response)

Reviewer #3: (No Response)

7. PLOS authors have the option to publish the peer review history of their article (what does this mean?). If published, this will include your full peer review and any attached files.

Reviewer #1: No

Reviewer #2: No

Reviewer #3: No

---

## [Editor Report · Acceptance letter]

28 Dec 2020

PONE-D-20-32590R1 

Performance Evaluation of survival regression models in analysing Swedish dental implant complication data with frailty 

Dear Dr. Fagbamigbe:

I'm pleased to inform you that your manuscript has been deemed suitable for publication in PLOS ONE. Congratulations! Your manuscript is now with our production department. 

Kind regards, 

on behalf of

Dr. Feng Chen 

Academic Editor

PLOS ONE